# Prospective Evaluation of Low-Dose External Beam Radiotherapy (LD-EBRT) for Painful Trapeziometacarpal Osteoarthritis (Rhizarthrosis) on Pain, Function, and Quality of Life to Calculate the Required Number of Patients for a Prospective Randomized Study

**DOI:** 10.3390/medsci9040066

**Published:** 2021-10-27

**Authors:** Robert Michael Hermann, Annika Trillmann, Jan-Niklas Becker, Alexander Kaltenborn, Mirko Nitsche, Mike Ruettermann

**Affiliations:** 1Center for Radiotherapy and Radiooncology Bremen and Westerstede, 26655 Westerstede, Germany; nitsche@strahlentherapie-nord.com; 2Department of Radiotherapy and Special Oncology, Hannover Medical School, 49511 Hannover, Germany; strahlentherapie@mh-hannover.de; 3Department of Anaesthesia, Federal Armed Forces Hospital Westerstede, 26655 Westerstede, Germany; Annika_Trillmann@web.de; 4Department of Trauma and Orthopaedic Surgery, Section for Plastic, Reconstructive and Hand Surgery, Federal Armed Forces Hospital Westerstede, 26655 Westerstede, Germany; alexander.kaltenborn@gmx.de (A.K.); mikeruettermann@yahoo.com (M.R.); 5Radiotherapy, Karl-Lennert-Krebscentrum, Universität Kiel, 24105 Kiel, Germany; 6HPC-Institute for Hand and Plastic Surgery, 26122 Oldenburg, Germany; 7University Medical Center Groningen, Department of Plastic Surgery, University of Groningen, 9713 Groningen, The Netherlands

**Keywords:** trapeziometacarpal osteoarthritis, thumb carpometacarpal, rhizarthrosis, low-dose radiotherapy, osteoarthritis, TMC arthrosis

## Abstract

*Background:* Retrospective studies have described the effectiveness of low-dose radiotherapy (LD-EBRT) in painful arthrosis of small finger joints, but two recent prospective studies have yielded ambiguous results. To generate accurate data for the planning of a trial, we conducted a prospective, monocentric, observational study to describe the effects of LD-EBRT as precisely as possible. *Methods*: Twenty-five consecutive patients with symptomatic trapeziometacarpal (TMC) arthrosis were irradiated with 6 × 0.5 Gy. Before, 3, and 12 months after LD-EBRT, we assessed subjective endpoints (modified “von-Pannewitz score”, 10-point visual analogue scale (VAS), “patient-rated wrist evaluation” (PRWE)), and objective measurements (“active range of motion” (AROM), Kapandji index, grip strength, pinch grip). *Results*: At 3/12 months, 80%/57% reported partial and 4%/18% complete remission according to the “von-Pannewitz” score. VAS “overall pain” significantly decreased from a median of seven (IQR 4) at baseline to three (IQR 6; *p* = 0.046) and to two (IQR 2; *p* = 0.013). Similar results were obtained for VAS “pain during exercise”, VAS “pain during daytime”, and VAS “function”. “PRWE overall score” was reduced from 0.5 at baseline (SD 0.19) to 0.36 (SD 0.24, *p* = 0.05) and to 0.27 (SD 0.18, *p* = 0.0009). We found no improvements of the objective endpoints (AROM, Kapandji, grip strength) except for flexion, which increased from 64° (SD 12°) at baseline to 73° (SD 9.7°, *p* = 0.046) at 12 months. *Conclusions:* We recommend the PRWE score as a useful endpoint for further studies for this indication. To prove a 15% superiority over sham irradiation, we calculated that 750 patients need to be prospectively randomized.

## 1. Introduction

Osteoarthritis is one of the most common diseases in elderly patients. The World Health Organization expects osteoarthritis to become the fourth leading cause of severe disability [1]. Concerning osteoarthritis of the small finger joints, a large Dutch study showed a prevalence of 75% of radiological signs in women between 60 and 70 [2]. However, according to the American College of Rheumatology (ACR) criteria, meaningful clinical diagnoses were found in only 7% of women in a comparable cohort in another large study [3].

There is robust evidence for the female gender, age over 50, ethnicity (lower prevalence in Chinese women), higher grip strength (in men, especially regarding trapeziometacarpal osteoarthritis (in the following called “TMC arthrosis”)), and joint injuries (review in [4]) as risk factors for the development of osteoarthritis in small finger joints.

EULAR (European League Against Rheumatism) compiled treatment strategies for TMC arthrosis and evaluated them according to evidence levels [5]. In addition to pain relief, therapy aims to maintain the mobility, stability, and strength of the joint complex that is decisive for hand function. Furthermore, comorbidities, personal attitudes, and patients’ wishes must be taken into account.

Conservative treatment options are the first therapeutic approach for early symptomatic arthrosis, usually within a multimodal therapy concept. Therefore, non-steroidal anti-inflammatory drugs (NSAIDs) [6], hand therapy [7], orthoses/splinting [8], or intra-articular injections with hyaluronic acid and corticoids [9] play essential roles. However, NSAID’s long-term use is associated with gastrointestinal bleeding, kidney failure, and other side effects.

Once the conservative therapy options have been exhausted, various surgical treatments are available for TMC arthrosis. Different techniques have been established. The best evidence points to simple trapeziectomy, but there is still controversy on whether to combine this with ligament reconstruction and/or tendon interposition. Different types of joint prostheses exist, with results not superior to trapeziectomy, except for quicker reduction of pain postoperatively but with more long-term issues. Another option is arthrodesis. In a recent Cochrane review, insufficient evidence was found due to the low quality of published randomized studies to define the most beneficial technique regarding pain reduction, functional outcome, or harms [10]. However, this review was withdrawn in 2017 from the Cochrane Library. Post-surgical complications, such as weakness, restriction of motion, or pain syndrome have been reported in up to 20% [11]. Thus, the surgical method with the least complications seems preferable.

The evaluation of other conservative treatment methods is essential to delay the time until surgery for some patients. In patients who have a high perioperative risk due to comorbidities or who are not willing to undergo surgery for other reasons, low-dose external beam radiotherapy (LD-EBRT) is an alternative therapeutic option. In Germany and Central Europe, LD-EBRT (formerly called “Reizbestrahlung“) for painful degenerative diseases of the joints and tendons has been widely adopted [12,13]. It has also been established in the therapy of painful TMC arthrosis [14,15]. Protocols comprise an LD-EBRT series of two to three fractions weekly with 0.5 or 1 Gy single dose up to a total dose of 3 to 6 Gy. The lifetime risk of fatal radiation-induced cancer is negligible, lower than 0.3 per 1000, as most patients are older than 50 years [16]. In a collective of 84 patients (101 joints) treated with LD-EBRT, 70% reported at least partial amelioration of symptoms up to one year after therapy [14].

However, the rationale of LD-EBRT in symptomatic osteoarthritis of the hands has been challenged recently. In a randomized trial, 56 patients received 6 × 1 Gy for hand osteoarthritis or sham irradiation [17]. Three months after therapy, no significant differences in pain relief, function, quality of life, and inflammation were reported. Nevertheless, these results must be viewed critically, as the statistical design of the study assumed using a power of 80% as a massive benefit from LD-EBRT, which ultimately led to the small number of recruited patients. Furthermore, 70% of the patients had suffered from symptoms for more than five years; most of them had arthrosis in many finger joints. Especially in these patient groups, such high effectiveness of LD-EBRT is unrealistic [18].

In another recent prospective randomized study, no correlation was found between applied radiation doses and therapeutic outcome (6 × 0.5 Gy vs 6 × 0.05 Gy) [19].

Our study aimed to prospectively describe the exact subjective and objective effects of LD-EBRT in a group of patients with “isolated” TMC arthrosis. We wanted to provide evidence on the efficacy of LD-EBRT regarding pain reduction, function, and quality of life after short- (3 months) and long-term (12 months) follow-up. This assessment of the clinical effects should allow a meaningful design of a randomized trial that ultimately might define the role of LD-EBRT in TMC arthrosis.

## 2. Materials and Methods

We conducted a monocentric, prospective, observational study. The ethics committee of the Medizinische Hochschule Hannover approved the study (Nr. 3358-2016). All patients gave written informed consent for participation.

We asked consecutive patients to participate in this investigation between 08/2016 and 10/2018. Inclusion criteria were clinical symptomatic and radiologically proven TMC arthrosis, age above 39 years, and informed consent for study participation. Exclusion criteria were age younger than 40 years, posttraumatic arthrosis of the respective joint, previous surgery on the TMC, other joints of the affected hand, or contralateral TMC with symptomatic degenerative alterations.

After consenting to participate, the baseline examination was carried out. Follow-up examinations were scheduled at 3 and 12 months after LD-EBRT or after the second series of LD-EBRT, respectively.

### 2.1. Endpoints

We asked patients for a general self-assessment of the “overall effectiveness” of the LD-EBRT at 3- and 12-month control examinations. For this, we rated the responses according to a slightly modified “von-Pannewitz score” as “no change”, “partial remission”, and “complete remission” (the original score additionally differentiated between “improved” and “significantly improved” [20]).

Furthermore, the self-assessment of pain was measured by a 10-point visual analogue scale (VAS). The following categories were assessed: pain during exercise, pain at night, pain during the daytime, pain at rest after stress, warm-up pain, and overall pain. An additional endpoint was the self-assessment of “overall function” by VAS.

As an independent and straightforward tool to measure the influence of the TMC symptoms on daily activities and function, we decided to use the standardized questionnaire “patient-rated wrist evaluation” (PRWE) [21]. This questionnaire is validated and consists of 5 questions about pain and 10 questions on function. The answers are compiled into three values: “PRWE overall”, “PRWE pain”, and “PRWE function”. Higher values represent worse symptoms. This questionnaire is much shorter and more straightforward than other tools such as “Disabilities of Arm, Shoulder, and Hand” (DASH: 38 questions) [22]. For ease of presentation, the achieved point value of the scores was divided by 100 and presented as numbers between 0 and 1.

As objective endpoints, we documented: (a) “active range of motion” (AROM) of the ipsilateral and, as “control”, the contralateral thumb TMC joint using a goniometer (adduction, abduction, flexion, and extension), and the Kapandji index (to evaluate the opposition movement of the thumb); (b) grip strength (Jamar^®^ dynamometer) and pinch grip (grip strength between thumb and index finger) with a pinch gauge meter.

### 2.2. LD-EBRT

Patients were treated twice weekly, with a single dose of 0.5 Gy amounting to a total dose of 3 Gy. LD-EBRT was administered as 6 MVX photons with a linear accelerator. The patient stood at the side of the treatment table on which the affected hand was positioned. One portal ensuring a source–skin (bolus) distance of 100 cm was applied from a 0° gantry position. The treatment portal was defined according to the reported pain location with a field size of at least 5 cm × 3 cm. It included the TMC and the metacarpophalangeal joint (MCP-1). We placed a 10 mm thick bolus material on the body surface to ensure a sufficient dose build-up in the anatomical structures. The field boundaries were marked on the skin with a waterproof pencil to simplify the positioning of the patients’ hand for further treatment fractions.

In the case of insufficient treatment response three months after LD-EBRT, the patients were offered a second series to a cumulative total dose of 6 Gy.

### 2.3. Statistical Methods

Patient demographics, baseline and follow-up data, and LD-EBRT-specific data, were prospectively collected in a study database. The collected data were assessed for normal distribution using the Shapiro–Wilks test. If this test is significant (*p* < 0.05), the data are not normally distributed. Normally distributed data were presented as mean and standard deviation (SD) and compared with the Student’s t-test. Not normally distributed data were presented as median and interquartile range (IQR) and analyzed with the non-parametric Mann–Whitney U Test. We compared musculoskeletal function of the irradiated and the contralateral hand at baseline. Values before irradiation were compared with the results after 3 and 12 months of follow-up. The Statistica version 10 software 23.0 (StatSoft Europe, Hamburg, Germany) was used to perform these statistical analyses.

We planned and proposed a randomized study to assess the effectiveness of LD-EBRT for this indication prospectively with our study results. The necessary number of cases was calculated with GPower 3.1 (according to [23]).

## 3. Results

We recruited thirty-two patients between August 2016 and October 2018. Seven patients had to be excluded from analysis: five refused re-presentation for the 3-month check-up, one refused further participation in the study after diagnosis of malignant disease, one did not complete LD-EBRT due to personal reasons.

The median age of the twenty-five included patients was 66 years; 68% were female. Most had already been treated with NSAIDs and other conservative therapies without lasting success. The median duration of symptoms before LD-EBRT was 12 months. Table 1 shows further details. Interestingly, several AROM measurements were significantly impaired on the affected side compared to the contralateral side (such as abduction, flexion, extension), as was the pinch grip (Table 2).

The mean field size of LD-EBRT was 31 cm²; 11 patients were treated with field sizes ≤ 6 cm × 4 cm. All patients completed LD-EBRT as scheduled.

All patients showed up for their three-month follow-up; we could evaluate 17 patients after 12 months.

Due to insufficient treatment response three months after LD-EBRT, eight patients were offered and received a second series to a cumulative total dose of 6 Gy. 

### 3.1. Subjective Amelioration of Symptoms (Von-Pannewitz Score, VAS)

Three months after LD-EBRT, 80% (n = 20) reported partial and 4% (n = 1) complete remission of symptoms according to the modified “von-Pannewitz” score. In contrast, 16% (n = 4) of the patients experienced “no change”. At 12 months after LD-EBRT, of the 16 patients that came for follow-up, 25% (n = 4) reported a “no change” situation, while 57% (n = 9) reported “partial”, and 18% (n = 3) “complete remission”.

After LD-EBRT, VAS “overall pain” significantly decreased from a median of 7 (IQR 4) at baseline to 3 (IQR 6; *p* = 0.046) at 3 months, and to 2 (IQR 2; *p* = 0.013) at 12 months (Figure 1). Similar results were obtained for VAS “pain during exercise”, being reduced from 8 (IQR 1) to 5.7 (3.4; n.s.) at 3 months, and to 5.1 (2.6, *p* = 0.025) at 12 months. Furthermore, VAS “pain during daytime” decreased significantly from 6 (IQR 5) at baseline to 0.5 (IQR 3; *p* = 0.016) at 3 months, and to 1 (IQR 5, *p* = 0.03) at 12 months.

The VAS “function” was reduced from 5.7 (SD 2.7) at baseline to 4.17 (SD 2.9; *p* = 0.036) after 3 months, and to 2 (IQR 6; *p* = 0.004) after 12 months.

However, no significant differences were found for VAS “pain at night”, “at rest”, and “warm-up pain” (data not shown).

### 3.2. Changes in QoL (PRWE)

“PRWE overall score” was reduced from 0.5 at baseline (SD 0.19) to 0.36 (SD 0.24, *p* = 0.05) at 3 months, and to 0.27 (SD 0.18, *p* = 0.0009) at 12 months (Figure 2). Both sub-scores were reduced during the course of time: “PRWE pain” from 0.54 (SD 0.19) to 0.32 (SD 0.22, *p* = 0.02) at 3 months, and to 0.37 (SD 0.23, *p* = 0.017) at 12 months, and “PRWE function” from 0.44 (SD 0.22) to 0.29 (SD 0.22, *p* = 0.03) and to 0.19 (SD 0.16, *p* = 0.0006).

### 3.3. Improvement of Motion (AROM, Kapandji)

Abduction at baseline was significantly less on the symptomatic side (65°, IQR 20°) compared to the contralateral side (76° (12°), *p* = 0.01). Although abduction improved after LD-EBRT after 3 months and 12 months (70° (12°), and 72° (14°)), this did not reach statistical significance.

Flexion increased after LD-EBRT during long-term follow-up: from 64° (SD 12°) at baseline to 65° (SD 11°, *p* = 0.88) at 3 months, and to 73° (SD 9.7, *p* = 0.046) at 12 months (Figure 3).

However, no significant differences in adduction and extension of the thumb were documented (data not shown). The Kapandji index (a tool to assess opposition movement of the thumb) was not improved after LD-EBRT (data not shown).

### 3.4. Changes in Grip Strength

The grip strength improved from 23.8 kg (SD 12 kg) to 24 kg (SD 11 kg) after 3 months and to 27 kg (SD 9 kg) after 12 months without reaching statistical significance. Similar results were obtained for pinch grip, which was reduced compared to the contralateral side at baseline (median 10 kPa (IQR 13) vs. 15 kPa (IQR 12, *p* = 0.003). However, the affected side’s pinch grip did not improve significantly during follow-up (data not shown).

## 4. Discussion

Our prospective study has limitations. Data were collected on paper without an electronic CRF, therefore we missed some data. Furthermore, there was an apparent “loss to follow-up” in the 12-month control, so that the database for the long-term observation was more uncertain than in the 3-month control. In addition, the sample size of 25 patients was relatively small. This is because recruitment was relatively slow. In order to generate the most meaningful results possible, we only included highly selected patients in our study (only TMC arthrosis of the hand, no other affected finger joints, no affected joints on the contralateral hand).

Still, the results fit well into previously reported results of LD-EBRT in comparable indications and fractionations (Table 3).

Two recent randomized trials on the efficacy of LD-EBRT for symptomatic arthrosis of the hand do not differentiate between TMC arthrosis and arthrosis of “small finger joints”. Taking the particular anatomy and function of the TMC into account, the applicability of the results is considerably limited.

(a) Minten et al. randomized 56 patients with osteoarthritis of the small finger joints between LD-EBRT with 6 × 1 Gy within two weeks and sham irradiation/“sound of LD-EBRT” [17]. The distal and proximal interphalangeal joints of fingers two to five were treated in both hands. These patients suffered pain NRS ≥ 5, despite NSAI or physiotherapy. Patients with the involvement of one hand only were excluded. Almost 80% of the included patients were female. The mean age was 65 years, and the mean BMI 27 kg/m². The primary study endpoint was the proportion of patients with a response three months after LD-EBRT. The response was defined as a relative improvement (at least 50%) and an absolute improvement (20%) in “pain” and “function”, or a relative improvement of 20% and an absolute improvement of 10% in two of the three items “pain”, “function”, and “PGA”. After three months, there were no significant differences in the proportion of patients with treatment responses after LD-EBRT (29% vs. 36% after sham). This corresponded to an odds ratio of 0.69 (95%CI 0.22–2.17). There were also no differences in objectively evaluated signs of inflammatory activity. The authors concluded that there was no substantial effect of LD-EBRT on symptoms and inflammatory signs in this clinical setting compared to sham. However, despite randomization, potential risk factors were unequally distributed between the two study arms. In the LD-EBRT group, the mean age was 67 years (vs. 63 years in sham), 68% had symptoms for at least five years (vs. 54%), 50% had erosive joint changes (vs 32%). Furthermore, only a relatively small number of patients was recruited. This was caused by the basic assumption of an enormous benefit of 40% in the proportion of responders by LD-EBRT, in addition to a 40% improvement in the placebo group. Thus, a moderate effect of LD-EBRT cannot be excluded for statistical reasons, even though such an effect is more realistic to achieve.

(b) The second randomized trial on the efficacy of LD-EBRT, ARTHRORAD, is just in the process of publishing [19]. One hundred and thirty-three patients were included, with 158 symptomatic osteoarthritis of the hands in addition to 63 painful knees. They were randomized between 6 × 0.5 Gy (standard dose) and 6 × 0.05 Gy (experimental dose, referred to as “very low dose”). The trial was closed prematurely due to slow recruitment before reaching the initially planned 270 patients. Mean age was about 67 years, with a mean duration of symptoms of over 4 years; a high percentage was treated on both hands. After 3 months, VAS was reduced by a mean of 18.9 points (SD 27.2) after a standard dose vs. 15.8 points (SD 25.5) after very low-dose LD-EBRT. The SF-SACRH score (for chronic rheumatic affections of the hands) increased from a mean of 23.1 (SD 10.6) to 74 (SD 5.7) after a standard dose in comparison to an increase from 20.7 (10.4) to 80 (4.4) after a very low dose. There were no significant differences in treatment outcomes after a standard and very low-dose LD-EBRT. In conclusion, this study failed to show a dose-dependent efficacy of LD-EBRT.

An explanation for this result could be that biological effects can be assumed already after 6 × 0.05 Gy, which may unfold as clinically similar to the effects after 6 × 0.5 Gy. In various cell models, precise changes in gene expression patterns for signal transduction, regulation of transcription, and metabolism were described in vitro and in vivo after radiation doses of 0.01 Gy or 0.04 Gy [27,28]. Epigenetic changes, such as an increase in DNA methylation levels have been shown in mice hematopoietic stem cells after 0.1 Gy, leading to a “low radiation-induced epigenetic reprogramming” [29]. This means that a single dose of 0.05 Gy must not be regarded as a “sham” or placebo when planning and interpreting this study.

In contrast to the above-cited randomized studies, we tried to recruit patients with symptomatic changes in the TMC in one hand without symptomatic arthrosis in other small finger joints, so that the effects on pain and function could be evaluated more clearly. Furthermore, our patients reported symptoms for about one year, much shorter than in the other two trials.

Four observational studies on treatment outcome after LD-EBRT of osteoarthritis of small finger joints have been published in the last 20 years (Table 3).

Keilholz et al. reported n = 19 patients with TMC arthrosis (20 joints) treated with X-ray radiation (120 kV) (12 × 1 Gy, three fractions per week, with a treatment break of 6 weeks in the middle of the series) [25]. The field size used was 6 × 8 cm. A subjective response 6 weeks after LD-EBRT was documented in 55% of the joints. A “thumb score” showed an improvement in 57%, especially in the criteria “pain” and “everyday functions”, and less for the items “mobility” and “special functions” [25,26]. The beneficial effects were also detectable in the long-term follow-up (median 4 years), although no exact data were given in either publication.

Kaltenborn et al. reported subjective treatment responses of 84 patients (101 irradiated TMC joints) after 6 × 1 Gy irradiation with 6 MVX photons [14]. After 3 months, 14% reported a complete and 49% a partial remission, and at 12 months 26%/44%. Interestingly, applying large treatment fields (> 6 × 4 cm) was a statistically significant independent factor for remission after LD-EBRT.

Hautmann et al. retrospectively evaluated the treatment results of n = 159 patients (with 295 irradiated joints), including n = 35 patients with TMC arthrosis [24]. LD-EBRT was completed with 6 MVX photons of a linear accelerator with 6 × 1 Gy (some patients received 6 × 0.5 Gy), three times per week. Pain intensity was reported by the patients using a numeric scale. The overall response across all irradiated joints was 45.7% at the end of LD-EBRT, 63.2% at 6 weeks, 65.7% at 12 weeks, and almost 65% in the long-term follow-up to 2 years later. For the subgroup of patients with TMC arthrosis, a significant reduction in pain during follow-up was reported (*p* < 0.001), without giving more precise data.

In 2020, the largest retrospective monocentric study of LD-EBRT in osteoarthritis of small finger joints to date was published. Donaubauer et al. reported on almost 500 patients [15]. In total, 49% (n = 236) of these patients were treated only at the thumb for TMC arthrosis, 26% (n = 127) additionally on other fingers, and 25% (n = 120) only on fingers 2–5. KV irradiation was used twice weekly with 6 × 0.5 Gy (5% of the collective with 6 × 1 Gy). A 6 cm × 8 cm field was used for the single involvement of TMC. In 85%, a second series was applied for an insufficient response after 6 weeks. Approximately 70% of the patients reported a reduction in symptoms of at least 20% within 24 weeks. About 30% showed no response, while 18% reported complete remission. In a direct comparison of the “involvement patterns”, patients with isolated TMC arthrosis reported a significantly worse therapy outcome than patients with involvement of both the thumb and other finger joints (*p* = 0.016). Patients achieved even better treatment results without “thenar involvement”, who had only received LD-EBRT for fingers 2–5 (*p* = 0.029). Nevertheless, in some patients with TMC arthrosis, LD-EBRT achieved complete symptom remission.

In the overall VAS assessment, data on symptomatic response in our collective correspond well with the results of the above-cited observational series. Overall, more than 50% report at least a partial remission of symptoms (as a surrogate for **relative** symptom reduction) within 3 to 6 months after LD-EBRT. However, up to 30% do not experience any relief. Compared to the randomized data, there is a 20 to 30% **absolute** reduction in VAS symptoms after LD-EBRT and after very low-dose treatment, as discussed above.


*What would be the general prerequisites for a successful prospective randomized trial?*


Adequate patient selection seems to be crucial. Inclusion of patients with a limited number of affected joints, optimally solitary symptomatic osteoarthritis of the TMC (or another single small finger joint), and select a collective with another prognosis rather than including patients with multiple affected small finger joints. Furthermore, only patients with symptomatic osteoarthritis that has not yet become chronic should be recruited, i.e., with a maximum medical history of 2 years.The control group should receive sham irradiation as performed in the Dutch study (“acoustic” irradiation) [17]. Since biological effects cannot be ruled out even with extremely low radiation doses, an effect of LD-EBRT cannot otherwise be disproved in case of comparable efficacy between the two study arms.Since the effect of LD-EBRT is not limited to the reduction in pain but also improves the subjective functionality of the hand, we propose the PRWE score as the primary endpoint, which captures both aspects well. We assume a 15% higher efficacy after LD-EBRT than sham irradiation, taking into account the undulating clinical course of symptoms in activated arthrosis.

Considering a “PRWE overall score” of 0.36 (SD 0.24) 3 months after LD-EBRT using the results from this study in comparison with a 15% higher score after sham irradiation, the sample size calculations revealed a necessary minimum study population of 312 patients to be recruited for each arm for a p-value of 0.05 and 80% power, respectively (analysis with GPower according to Faul [23], Table 4). To compensate for loss to follow-up of about 20% of patients after LD-EBRT, a minimum patient population of 375 patients per arm is recommended.

However, such a large study with 750 patients in this setting is only conceivable if a sufficiently large number of study centers participate. This requires a straightforward study protocol with clear inclusion criteria, central randomization, and clear endpoints that are easy to collect. The PRWE questionnaire is ideally suited, as we showed in our study.

Would such a large-scale trial of this clinical endpoint be clinically relevant?

Yes, since LD-EBRT could be secured as a therapeutic option to delay the necessity of surgical interventions for the TMC joint without the systemic side effects of NSAIDs and at similarly low costs as complex or repeatedly adjusted splinting. Furthermore, patients who refuse surgery or who for functional reasons cannot use a splint at work, for example, might benefit from this additional non-surgical treatment option.

## Figures and Tables

**Figure 1 medsci-09-00066-f001:**
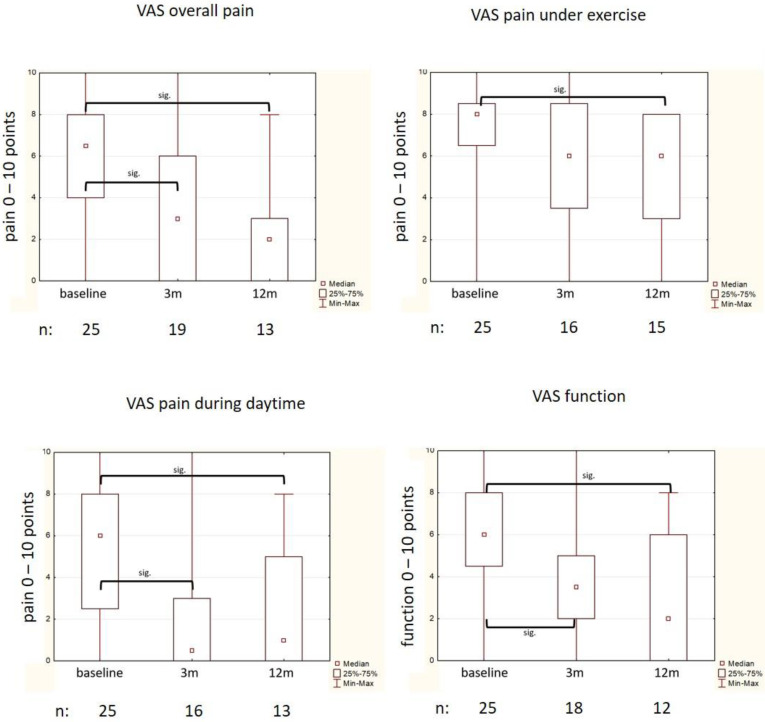
VAS values before LD-EBRT (baseline) and after 3 mo and 12 mo after LD-EBRT (median, IQR, min/max).

**Figure 2 medsci-09-00066-f002:**
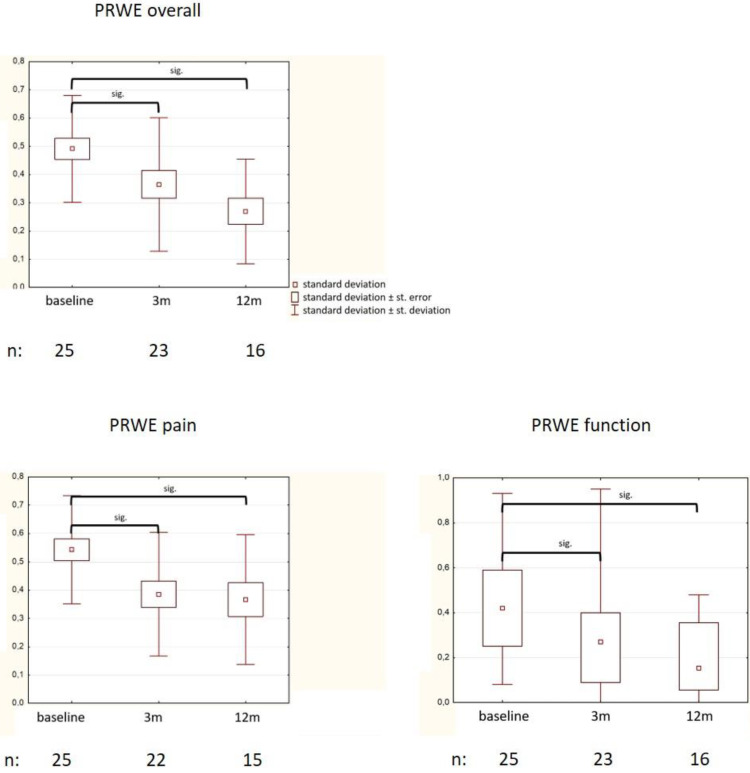
Changes in PRWE (patient-rated wrist evaluation) values (mean value, standard error, standard deviation are given at each time point).

**Figure 3 medsci-09-00066-f003:**
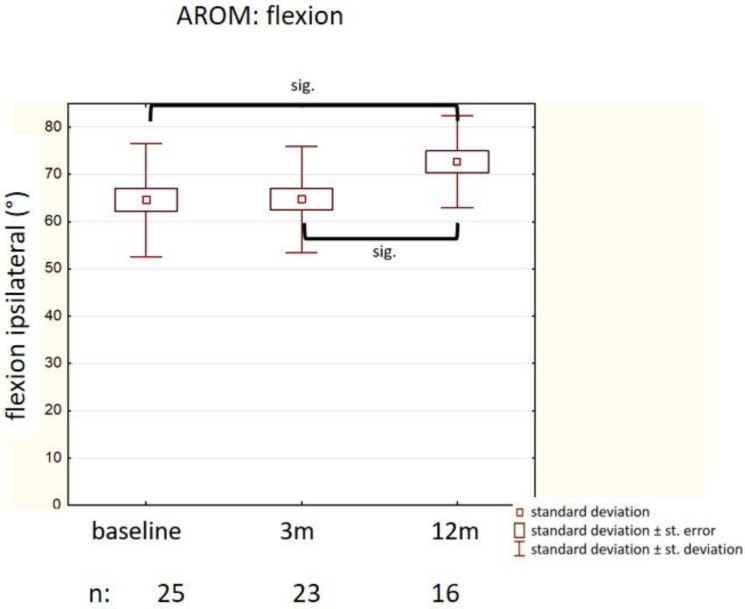
Improvement in flexion of the thumb (mean value, standard error, standard deviations are given at each time point).

**Table 1 medsci-09-00066-t001:** Patient characteristics.

	Mean */Median# (SD */IQR#)	Min/Max
Age	# 66 (18)	47/80
Sex:		
Female	17 (68%)	
Male	8 (32%)	
BMI *	* 27.29 (4.14)	19/35.5
Previous medication:		
None	9	
NSAI	14	
Steroids	2	
Previous conservative therapies:		
Physiotherapy	4	
Orthesis/splints	10	
Injections	4	
Duration of symptoms before irradiation	# 12 (20)	1/240
<3 months	4	
3–12 months	11	
>12 months	10	
VAS pain		
- Overall pain	# 7 (4)	0/10
- Under exercise	# 8 (1)	0/10
- At night	# 3 (6)	0/8
- During day	# 6 (5)	0/10
- At rest	# 5 (6)	0/9
- Warm-up pain	# 0 (6)	0/8
VAS function	* 5.76 (2.66)	0/10
PRWE pain	* 0.54 (0.19)	0/0.8
PRWE function	* 0.44 (0.22)	0.08/0.93
PRWE total	* 0.5 (0.19)	0.05/0.82

* mean, SD; # median, IQR; VAS: visual analogue scale; PRWE: patient-rated wrist evaluation.

**Table 2 medsci-09-00066-t002:** Patient characteristics: AROM before LD-EBRT.

AROM	Mean/Median (SD/IQR)	Min/Max	
Adduction (ipsilateral) (cm)	# 1.5 (0.8)	0.5/2.5	# *p* = 0.62
Adduction (contralateral) (cm)	* 1.16 (0.56)	0.3/2.5	
Abduction (ipsilateral) (°)	# 65 (20)	40/85	* *p* = 0.01
Abduction (contralateral) (°)	* 76.76 (11.9)	55/110	
Flexion (ipsilateral) (°)	* 64.6 (11.9)	40/90	* *p* = 0.0003
Flexion (contralateral) (°)	* 77.8 (11.7)	50/100	
Extension (ipsilateral) (°)	# 8 (5)	0/20	# *p* = 0.003
Extension (contralateral) (°)	# 10 (5)	5/40	
Kapandji (ipsilateral)	# 7.5 (2.5)	4/9	# *p* = 0.13
Kapandji (contralateral)	# 8.5 (2)	4/10	
Grip strength (ipsilateral) (kg)	* 23.8 (11.9)	5/46	# *p* = 0.07
Grip strength (contralateral) (kg)	# 25 (18)	10/52	
Pinch grip (ipsilateral) (kPa)	# 10 (13)	0/30	# *p* = 0.003
Pinch grip (contralateral) (kPa)	# 15 (12)	2/32	

* mean, SD, *t*-test; # median, IQR, MWU; LD-EBRT: low-dose external beam radiotherapy; AROM: active range of motion.

**Table 3 medsci-09-00066-t003:** Comparison of own data with contemporary studies on LD-EBRT of rhizarthrosis/arthrosis of small finger joints. kV: kilovolts (“X-rays”), MVX: megavolts (“linear accelerator radiation”), LD-EBRT: low-dose external beam radiotherapy.

	N	Technique	Dose	Results	Commentary
*randomized studies*					
Minten [17]	56	MVX	6 × 1 Gy 3 × per week 6 × sham	Response at 3 mo: 29% (LD-EBRT) vs. 36% (n.s.)	No TMC arthrosis
ARTHRORAD [19]	158 hands	MVX	6 × 0.5 Gy 2 × per week 6 × 0.05 Gy	VAS reduction at 3 m: 18.9 points (SD 27.2) vs. 15.8 (SD 25.5) (n.s.)	Small finger joints, no exclusive recruitment of TMC arthrosis
*observational studies*					
Hautmann [24]	35 (159 finger joints)	MVX	6 × 1 Gy (6 × 0.5 Gy) 3 × per week	All patients 46% response during LD-EBRT, 65% within 2 years	No subgroup analysis for TMC
Donaubauer [15]	236 + 127	kV	6 × 0.5 Gy (6 × 1 Gy) 2 × per week	70% at least 20% response within 6 months Worse response in case of TMC arthrosis	Response dependent on involvement patterns of small finger joints
Keilholz [25,26]	19 (20)	kV	12 × 1 Gy split course 3 × per week	6 weeks after LD-EBRT: 55% response in pain	All TMC arthrosis
Kaltenborn [14]	84 (101)	MVX	6 × 1 Gy 2 × per week	70% at least partial response up to 1 year (30% complete response)	All TMC arthrosis
Own data	25	MVX	6 × 0.5 Gy 2 × per week	Response at 3 mo: VAS pain sig. reduced from 7 (IQR 4) to 3 (IQR 6) PRWE pain from 0.54 (SD 0.19) to 0.32 (SD 0.22, *p* = 0.02) PRWE function from 0.44 (SD 0.22) to 0.29 (SD 0.22, *p* = 0.03)	

**Table 4 medsci-09-00066-t004:** Sample size calculations with GPower 3.1.9.7.

T Tests	Means: Difference between Two Independent Means (Two Groups)	
Analysis	A priori: Compute required sample size	
**Input:**	**Tails**	**Two**
	Effect size d	0.2250000
	α error	0.05
	Power (1-β err prob)	0.8
	Allocation ratio N2/N1	1
**Output:**	Noncentrality parameter δ	2.8102491
	Critical t	1.9637852
	Df	622
	Sample size group 1	312
	Sample size group 2	312
	Total sample size	624
	Actual power	0.8012088

Assumption: considering a “PRWE overall score” of 0.36 (SD 0.24) within the first 3 months after LD-EBRT in comparison with a 15% higher score after sham irradiation.

## Data Availability

The raw data generated in this study can be obtained from the corresponding author on request.

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
