# Peer review of "Prospective Evaluation of Low-Dose External Beam Radiotherapy (LD-EBRT) for Painful Trapeziometacarpal Osteoarthritis (Rhizarthrosis) on Pain, Function, and Quality of Life to Calculate the Required Number of Patients for a Prospective Randomized Study"

_medsci, 2021, doi:10.3390/medsci9040066_

Round 1
Reviewer 1 Report
I believe that this paper contains groundbreaking data that could greatlybenefit for people with osteoarthritis. But this data are very preliminary.
It is necessary include in exclusion criteria farmacological therapy. Have the authors evaluated the therapy of the single partecipant?
I think that this study can be published underlining that these are
preliminary data. In the next study, instead of significantly increasing the participants,
I would make restrictions, so I would consider subjects with osteoarthritis in
the same hand, only in the hand, and with a comparable level of use and comparable functional limitation.
Author Response
First of all, we want to thank the reviewers for the thoughtful and helpful comments.
In the following, we discuss the comments point by point.
Reviewer 1
"I believe that this paper contains groundbreaking data that could greatly benefit for people with osteoarthritis. But this data are very preliminary.
It is necessary include in exclusion criteria farmacological therapy. Have the authors evaluated the therapy of the single participant?"
- Answer: In Germany, it is clinical practice for most patients with pain due to joint osteoarthritis to be prescribed NSAIDs by their general practitioners. Therefore, in order to be able to recruit a substantial number of patients for our collective, we could not define "pharmacological therapy" as an exclusion criterion. In total, 14 patients already received NSAIDs at the first presentation in our practice, whereas nine did not. At the clinical follow-up appointments, we asked the patients whether they had started taking NSAIDs (or other therapies) NEW in the meantime. All patients declined this. In this respect, the effects measured in the study can be attributed to the radiotherapy and not to a newly started NSAI intake.
"I think that this study can be published underlining that these are preliminary data. In the next study, instead of significantly increasing the participants, I would make restrictions, so I would consider subjects with osteoarthritis in the same hand, only in the hand, and with a comparable level of use and comparable functional limitation."
> Answer: The reviewer's suggestion is excellent. However, we had already made a clear selection when planning our study: Thus, only patients with clinically symptomatic, radiologically proven TMC osteoarthritis in one hand were recruited. Furthermore, the other finger joints were not allowed to be affected. Although we receive about 100 patients with activated arthrosis for radiotherapy per month, it still took two years to recruit such a collective - because of the strict inclusion- and exclusion criteria. We fear that the additional consideration of pre-existing functional limitations will make the recruitment of a sufficiently large collective almost impossible, even in a multicentre study.
"Preliminary data:"
> Answer: We agree that our research cannot scientifically prove the effectiveness of radiotherapy in this indication. In this respect, our data are "preliminary". However, the aim of our investigation was different: we wanted to collect data that enabled us to plan a study that could provide scientific proof.
We have added a paragraph in the discussion section (p. 9): ““In addition, the sample size of 25 patients was relatively small. This is because recruitment was relatively slow. In order to generate the most meaningful results possible, we only included highly selected patients in our study (only TMC arthrosis of the hand, no other affected finger joints, no affected joints on the contralateral hand).”
Reviewer 2 Report
Dear authors,
Thank you for providing the article entitled "Prospective Evaluation of Low-Dose External Beam Radiotherapy (LD-EBRT) for Painful Trapeziometacarpal Osteoarthritis (Rhizarthrosis) on Pain, Function, and Quality of Life to Calculate the Required Number of Patients for a Prospective Randomized Study". It is an interesting article and well-conducted research on the role of LD-EBRT in the management of enthesopathies. There are some comments to improve the text and increase its visibility:
1. Please see this comprehensive review article in this context and use it as a citation if it meets your criteria.
- Seyed Alireza Javadinia, Nooshin Nazeminezhad, Ruhollah Ghahramani-Asl, Davood Soroosh, Danial Fazilat-Panah, Babak PeyroShabany, Seyedeh Naeimeh Saberhosseini, Arezoo Mehrabian, Farzad Taghizadeh-Hesary, Mohammad Nematshahi, Gaurav Dhawan, James S. Welsh, Edward J. Calabrese & Rachna Kapoor (2021) Low-dose radiation therapy for osteoarthritis and enthesopathies: a review of current data, International Journal of Radiation Biology, 97:10, 1352-1367, DOI: 10.1080/09553002.2021.1956000
Moreover, there are several Dutch randomized trial in this context that should be cited properly and discussed in the the discussion:
- Mahler EAM, Minten MJ, Leseman-Hoogenboom MM, Poortmans PMP, Leer JWH, Boks SS, van den Ende CHM. 2019. Effectiveness of low-dose radiation therapy on symptoms in patients with knee osteoarthritis: a randomised, double-blinded, sham-controlled trial. Ann Rheum Dis. 78(1):83–90.
- Minten MJM, Leseman-Hoogenboom MM, Kloppenburg M, Kortekaas MC, Leer JW, Poortmans PMP, van den Hoogen FHJ, den Broeder AA, van den Ende CHM. 2018. Lack of beneficial effects of low-dose radiation therapy on hand osteoarthritis symptoms and inflammation: a randomised, blinded, sham-controlled trial. Osteoarthr Cartil. 26(10):1283–1290.
- van den Ende CHM, Minten MJM, Leseman-Hoogenboom MM, van den Hoogen FHJ, den Broeder AA, Mahler EAM, Poortmans PMP. 2020. Long-term efficacy of low-dose radiation therapy on symptoms in patients with knee and hand osteoarthritis: follow-up results of two parallel randomised, sham-controlled trials. Lancet Rheumatol. 2(1):e42–e49.
2. The sample size is too small. Please state it as one of limitations of your study.
Author Response
First of all, we want to thank the reviewers for their thoughtful and helpful comments.
In the following, we discuss the comments point by point.
- Please see this comprehensive review article in this context and use it as a citation if it meets your criteria: Seyed Alireza Javadinia, Nooshin Nazeminezhad, Ruhollah Ghahramani-Asl, Davood Soroosh, Danial Fazilat-Panah, Babak PeyroShabany, Seyedeh Naeimeh Saberhosseini, Arezoo Mehrabian, Farzad Taghizadeh-Hesary, Mohammad Nematshahi, Gaurav Dhawan, James S. Welsh, Edward J. Calabrese & Rachna Kapoor (2021) Low-dose radiation therapy for osteoarthritis and enthesopathies: a review of current data, International Journal of Radiation Biology, 97:10, 1352-1367, DOI: 10.1080/09553002.2021.1956000
> Answer: Thank you for the excellent advice. This citation fits very well in our introduction (citation 13).
Moreover, there are several Dutch randomized trial in this context that should be cited properly and discussed in the the discussion:
- Mahler EAM, Minten MJ, Leseman-Hoogenboom MM, Poortmans PMP, Leer JWH, Boks SS, van den Ende CHM. 2019. Effectiveness of low-dose radiation therapy on symptoms in patients with knee osteoarthritis: a randomised, double-blinded, sham-controlled trial. Ann Rheum Dis. 78(1):83–90.
> Answer: We had refrained from a detailed presentation and citation of this study, as only patients with osteoarthritis of the KNEE were included and examined in this study. Our study focused on osteoarthritis of the TMC joint (only one isolated joint in the HAND). If we presented and discussed data on other joints/anatomical regions, we would overburden our rather long and extensive discussion. Discussing other joints would also dilute the goal of our study, which is to generate data for a prospective randomized study on TMC arthrosis, which would then also have the chance to prove or reject the effect of LD-EBRT scientifically.
- Minten MJM, Leseman-Hoogenboom MM, Kloppenburg M, Kortekaas MC, Leer JW, Poortmans PMP, van den Hoogen FHJ, den Broeder AA, van den Ende CHM. 2018. Lack of beneficial effects of low-dose radiation therapy on hand osteoarthritis symptoms and inflammation: a randomised, blinded, sham-controlled trial. Osteoarthr Cartil. 26(10):1283–1290.
> Answer: The proposed study has already been cited in the manuscript (initial citation 16, after revision now citation 17). In addition, the study is described and criticized in detail in the discussion (page 9, but also Table 3).
- van den Ende CHM, Minten MJM, Leseman-Hoogenboom MM, van den Hoogen FHJ, den Broeder AA, Mahler EAM, Poortmans PMP. 2020. Long-term efficacy of low-dose radiation therapy on symptoms in patients with knee and hand osteoarthritis: follow-up results of two parallel randomised, sham-controlled trials. Lancet Rheumatol. 2(1):e42–e49.
> Answer: Since this citation is only a follow-up of the "Minten study" already described above, mixed with the follow-up of a study on knee osteoarthritis, we have refrained from citing these data ourselves. In [17], the data of the "Minten study" that are decisive from our point of view have been reported - and have already been presented and criticized in detail in our discussion.
- The sample size is too small. Please state it as one of limitations of your study.
> Answer: We have added to the discussion accordingly (at the beginning of the section, on p. 9):
“In addition, the sample size of 25 patients was relatively small. This is because recruitment was relatively slow. In order to generate the most meaningful results possible, we only included highly selected patients in our study (only TMC arthrosis of the hand, no other affected finger joints, no affected joints on the contralateral hand).”